# PLA2R1 Inhibits Differentiated Thyroid Cancer Proliferation and Migration via the FN1-Mediated ITGB1/FAK Axis

**DOI:** 10.3390/cancers15102720

**Published:** 2023-05-11

**Authors:** Hui Zheng, Mengyu Zhang, Dingwei Gao, Xiaoying Zhang, Haidong Cai, Zhijun Cui, Yang Gao, Zhongwei Lv

**Affiliations:** 1Department of Nuclear Medicine, Shanghai Tenth People’s Hospital, Tongji University School of Medicine, Shanghai 200092, China; 2Department of Medicine Imaging, the Chongming Branch of Shanghai Tenth People’s Hospital, Tongji University, Shanghai 200092, China; 3Department of Neurosurgery, Fudan University Shanghai Cancer Center, Shanghai 200032, China; 4Department of Oncology, Shanghai Medical College, Fudan University, Shanghai 200032, China

**Keywords:** PLA2R1, thyroid cancer, FN1, ITGB1/FAK, migration

## Abstract

**Simple Summary:**

PLA2R1 is a promising biological candidate for exploitation in thyroid cancer. However, the mechanism of action of PLA2R1 in thyroid cancer has not been fully elucidated. We found that PLA2R1 was expressed at low levels in thyroid cancer tissues and that its expression level was positively correlated with prognosis. PLA2R1 competed with FN1 to bind ITGB1 to mediate extracellular matrix remodeling and thereby inhibit thyroid cancer progression. We investigated the mechanism of PLA2R1 in thyroid cancer and provided a new strategy for thyroid cancer treatment that targets PLA2R1.

**Abstract:**

PLA2R1 is a novel gene that is aberrantly expressed in a variety of malignancies. However, the role and mechanism of PLA2R1 in thyroid cancer has not been elucidated. We aimed to uncover the underlying mechanism of PLA2R1 in thyroid cancer. We collected 115 clinical specimens, including 54 tumor tissues and 61 para-cancerous tissues, who underwent surgical treatment at Shanghai Tenth Hospital. Immunohistochemical staining was used to evaluate PLA2R1 expression in differentiated thyroid cancer (DTC) tissues. The thyroid cancer cell lines 8505c and FTC133 transfected with PLA2R1 overexpression or knockdown plasmids were used for CCK8 assays and a wound healing assay. Next, we conducted coimmunoprecipitation (Co-IP) experiments and western blotting to explore the underlying mechanism of PLA2R1 in regulating the growth of thyroid cancer. We discovered that the expression of PLA2R1 was lower in the tumor tissues than in para-cancerous tissues (χ^2^ = 37.0, *p* < 0.01). The overexpression of PLA2R1 significantly suppressed thyroid cancer cell proliferation and migration, and both of these effects were partially attenuated by the knockdown of PLA2R1. Furthermore, the in vivo growth of DTC could be alleviated by the knockdown of PLA2R1. The mechanistic study revealed that PLA2R1 competed with FN1 for binding to ITGB1, inhibiting the FAK axis and epithelial-mesenchymal transition (EMT). We speculate that PLA2R1 might be a promising marker and a novel therapeutic target for thyroid cancer.

## 1. Introduction

Thyroid cancer is the most common endocrine cancer, with an increasing incidence over the past 30 years [1,2]. According to cancer statistics, in 2018, thyroid cancer had the 9th highest incidence rate worldwide, with a morbidity incidence of 10.2 per 100,000 person-years [3]. Thyroid cancer includes a number of subtypes, such as follicular, papillary, anaplastic and medullary thyroid carcinoma. Differentiated thyroid cancer (DTC), comprising mainly papillary thyroid carcinoma (PTC) and follicular thyroid carcinoma (FTC), is the most common subtype of thyroid cancer. The elevated incidence of thyroid cancer is very strongly related to the increased diagnosis of DTC. Moreover, it was reported that 20% of patients with DTC developed distant metastases or recurrence [4]. Under these conditions, conventional therapeutic approaches have several limitations in the treatment of thyroid cancer, and the survival time of patients can be significantly reduced [5]. Thus, it is important to identify novel prognostic factors and investigate the mechanism underlying the distant metastases of DTC.

The Cancer Genome Atlas (TCGA) database integrates data from all human cancers for systematic analysis to find all small variants of oncogenes to understand the mechanisms of cancer cell development and progression [6]. In this study, the expression profile of thyroid cancer was downloaded from TCGA. We found that the phospholipase A2 receptor 1 (PLA2R1) was expressed at low levels in thyroid cancer and that low PLA2R1 expression was associated with poor prognosis.

PLA2R1, a type I transmembrane receptor, is a member of the mannose receptor family and exists in transmembrane and soluble forms [7]. Belonging to the superfamily of phospholipases, PLA2R1 can catalyze the hydrolysis of phospholipids and then produce free lysophospholipids and fatty acids [8]. Recently PLA2R1 was found to regulate cell proliferation, growth, differentiation and apoptosis through a cascade of downstream signaling pathways [9,10]. In vitro studies have shown that PLA2R1 mRNA expression was reduced in several types of cancer, such as leukemia and kidney, thyroid and breast cancers [8,11,12,13]. Notably, Hosseinkhan et al. reported that PLA2R1 was downregulated in stage I and IV thyroid cancer and had a potential positive correlation with prognosis in intermediate-risk thyroid cancer [14,15]. The above findings suggest that PLA2R1 is a promising prognostic marker in thyroid cancer. However, the underlying mechanism of PLA2R1 in DTC has not been fully elucidated.

To further examine the function and clinical significance of PLA2R1, a series of in vitro assays and in vivo assays were conducted to investigate its action in the development of DTC. In addition, coimmunoprecipitation (Co-IP) and Kyoto Encyclopedia of Genes and Genomes (KEGG) analyses were utilized to further elucidate the underlying mechanism of PLA2R1 in DTC. It can be speculated that PLA2R1 is a potential therapeutic and prognostic biomarker for thyroid cancer.

## 2. Materials and Methods

### 2.1. Bioinformatics Analysis

The raw gene expression profiling data of DTC were assessed and downloaded from the TCGA on 12 April 2021 (https://portal.gdc.cancer.gov/) database to assess the gene expression of PLA2R1. To identify differentially expressed genes (DEGs), three R packages were applied to analyze the DTC expression data in the TCGA database. We chose the ten most upregulated and downregulated genes and then took the intersection of the results obtained with the three R packages. The cut-off criteria for DEGs were *p* < 0.05 and log2 FC (fold-change) > 1.0. The associations of overall survival (OS), progression-free interval (PFI) and disease-free survival (DFS) with PLA1R1 expression were assessed by using the Kaplan–Meier method with the log-rank test.

### 2.2. Tissue Samples

To further validate the expression of PLA2R1 in thyroid cancer tissues, a total of 54 DTC tissues and 61 para-cancerous tissues from Shanghai Tenth People’s Hospital, Tongji University School of Medicine, were evaluated. Written informed consent was obtained from all patients prior to their participation in this study. All tissues were used for immunohistochemical staining to analyze PLA2R1 expression. The selection criteria for the thyroid carcinoma patients included the following: (i) clinically diagnosed with DTC; (ii) aged between 18 and 75 years, without a history of other malignancies; (iii) undergoing surgical treatment. The exclusion criteria included the following: (i) patients not pathologically confirmed to have DTC following surgery; (ii) patients with any other severe mental or physical diseases. Sample collection was approved by the Ethics Committee of Shanghai Tenth People’s Hospital, Tongji University (see Table 1).

### 2.3. Immunohistochemical Staining

Tissue samples were sliced into 4 mm-thick sections for immunohistochemical (IHC) analysis. Then, IHC staining was performed using the standard immunoperoxidase staining procedure. Tissues were fixed in 10% formaldehyde in PBS, embedded in paraffin, sectioned and subjected to IHC staining with antibodies specific for human proteins. The specific antibody and concentration were, respectively, as follows: anti-PLA2R1, 1:1000, (Abcam, Cambridge, UK).

### 2.4. Cell Culture

The thyroid cancer cell lines BC-PAP, TPC-1, 8505c and FTC133, and the normal thyroid cell line NTHY-ORI3-1 were purchased from Gibco (Thermo, Waltham, MA, USA). The thyroid cancer cell lines BC-PAP, TPC1, and 8505c were cultured with Dulbecco’s modified Eagle’s medium (DMEM) (Thermo, Gibco, MA, USA), and supplemented with 10% fetal bovine serum (FBS) (Thermo, Gibco, MA, USA) and 1% penicillin–streptomycin (Thermo, Gibco, MA, USA). The thyroid cancer cell line FTC133 and normal thyroid cell line NTHY-ORI3-1 were maintained in Roswell Park Memorial Institute (RPMI)-1640 medium containing 10% FBS (Thermo, Gibco, MA, USA) and 1% penicillin–streptomycin (Thermo, Gibco, MA, USA). All cells were cultured at 95% humidity and 5% CO_2_ at 37 °C.

### 2.5. Plasmid Synthesis and Transfection

The PLA2R1 expression plasmid was synthesized by Yazai Biotechnology (Shanghai, China). The target sequences of the PLA2R1 overexpression plasmids were PLA2R1-F: 5′-AGACTCGAGGCCACCATGCTGCTGTCGCCGTCGCTGC-3′ and PLA2R1-R: 5′-GTCACTTAAGCTTTTTTGGTCACTCTTCTCAAGATCA-3. The target sequences of the shRNAs were as follows: PLA2R1-shRNA1-F: GCAGAAGTAGGTTGTGATACTCTCGAGAGTATCACAACCTACTTCTGCTTTTT, PLA2R1-shRNA1-R: AAAAAGCAGAAGTAGGTTGTGATACTCTCGAGAGTATCACAACCTACTTCTGC, PLA2R1-shRNA2-F: GGATTGGTTTGAGCAGCAATACTCGAGTATTGCTGCTCAAACCAATCCTTTTT, PLA2R1-shRNA2-R: AAAAAGGATTGGTTTGAGCAGCAATACTCGAGTATTGCTGCTCAAACCAATCC, PLA2R1-shRNA3-F: GCAAGAAACTGTGCTGTTTATCTCGAGATAAACAGCACAGTTTCTTGCTTTTT and PLA2R1-shRNA3-R: AAAAAGCAAGAAACTGTGCTGTTTATCTCGAGATAAACAGCACAGTTTCTTGC. The cell lines were then screened with puromycin to confirm stable transfection. The expression of PLA2R1 in thyroid cancer cell lines was determined by RT–PCR and Western blotting.

### 2.6. Real-Time Polymerase Chain Reaction (RT–PCR)

The total mRNA of different cell lines was extracted using Invitrogen products (Thermo, Gibco, MA, USA) according to the manufacturer’s specifications. The TransScript All-in-One First-Strand cDNA Synthesis SuperMix for RT–PCR kit (TransGen Biotech, Beijing, China) was used for reverse transcription. The primer sequences for RT–PCR are presented in Appendix A. The relative expression levels were calculated using GAPDH as the standard and ΔΔCt method.

### 2.7. Western Blot

The total protein from each sample was extracted with an RIPA buffer (Beyotime, Shanghai, China) for 30 min. Cell lysates were centrifuged at 14,000× *g* and 4 °C for 15 min. The supernatant containing the proteins was retained. The concentrations of proteins were then identified using the BCA Protein Assay Kit (Thermo Scientific, MA, USA). Protein samples (30 μg) were loaded onto 10% SDS–PAGE gels and transferred onto PVDF membranes. A quantity of 3 mL of 5% skim milk powder sealer was added to the antibody incubator. The membrane was sealed on a shaker for 1 h at room temperature and washed 3 times with TBST solution for 10 min each. The primary antibody (1:1000) was added and incubated overnight at 4 °C. The membrane was washed 3 times with TBST solution for 10 min each time. Then, HRP-labelled secondary antibody (1:2000) was added and incubated for 1 h at room temperature. The membrane was washed 3 times with the TBST solution for 10 min each time. The major antibodies used in Western blotting are presented in Appendix A. The intensity of the GAPDH band was then taken as the control for all other bands. Each test was repeated three times.

### 2.8. CCK-8 Assay to Evaluate Cell Proliferation

For the cell viability, after cells were transfected, the number of cells in the 96-well plates was counted using an inverted fluorescence microscope (Olympus Corporation, TKY, Tokyo, Japan) for 0–4 consecutive days. The cell proliferation assays were undertaken by the CCK-8 method. Starting from Day 1, 10 µL of CCK-8 reagent (BBI Life Sciences, Shanghai, China) solution was added to each well and incubated for 1 h. The absorbance at 450 nm was then measured with a microplate reader (BioTek, Winooski, VT, USA).

### 2.9. Wound Healing Assay

To assess the motility of the cells, a wound healing assay was carried out. Cells were inoculated in 6-well culture plates containing DMEM (Thermo, Gibco, MA, USA) and 10% FBS. Then, the confluent cells were scratched with the tip of a 10 µL pipette and washed three times with PBS to remove cell debris. After 48 h of incubation, cells migrating into the wounded area or cells protruding from the border of the wound were visualized and photographed.

### 2.10. Subcutaneous Xenograft Mouse Model

Nude mice (5–8 weeks, 18–20 g) were purchased from Shanghai SLAC Laboratory Animal company (Shanghai, China). All nude mice were raised and maintained in an SPF (Specific Pathogen Free) environment. After one week of adaptive feeding, an im-plantation site was selected in the axilla with an abundant blood supply. After thyroid cancer cells FTC133 were infected with shRNAs successfully, 5 × 10^6^ cells in 100 μL PBS were injected into the axilla of nude mice. Then, after inoculation of the thyroid cancer cells, tumor growth was closely monitored in each group. The longest and shortest part of the tumor were measured daily, and the tumor volume (mm^3^) was calculated using the following formula: 1/2  ×  a ×  b^2^ (a is the long axis; b is the short axis). At the end, the tumor weight (mg) was documented after the mice were euthanized. Tumor tissue was stored in liquid nitrogen for subsequent experiments and analysis. All animal experiments were approved by the Experimental Animal Ethics Committee of Shanghai Tenth People’s Hospital.

### 2.11. Coimmunoprecipitation (Co-IP) Assay

To explore the mechanism underlying the regulatory effects of PLA2R1 on thyroid cancer, RNA sequencing was performed to identify DEGs between normal thyroid tissues or cells and thyroid cancer tissues or cells. The selected DEGs were entered into the Gene Expression Profiling Interactive Analysis (GEPIA) database to verify the correlations between the corresponding proteins.

The proteins encoded by the selected DEGs were analyzed by a Co-IP experiment. Cell lysates were incubated with an anti-ITGB1 antibody (Abcam, Cambridge, UK) or control IgG (Cell Signaling Technology, BSN, Danvers, MA, USA) for 2 h, and 20 μL of Protein A/G PLUS-Agarose beads was then added and incubated overnight at 4 °C. The antibody-antigen conjugates were then washed with 500 μL of complete Co-IP/wash buffer, and the proteins were eluted in Laemmli sample buffer and denatured at 68 °C for 10 min. The eluates were analyzed by Western blotting.

### 2.12. Statistical Analysis

All data analyses were conducted using SPSS 22.0 software (SPSS Inc., Chicago, IL, USA). Data are shown as mean ± SD. A sign test was conducted to compare the gene expression levels of PLA2R1 between DTC and para-cancerous tissues.

Spearman rank correlation analysis and a two-tailed *t* test were applied to investigate whether the DEGs were significantly related to clinical and pathological parameters. Kaplan–Meier analysis was conducted to analyze survival rates. The log-rank test and χ^2^ test were carried out to compare differences. Differences were considered statistically significant at *p* < 0.05.

## 3. Results

### 3.1. The Expression of PLA2R1 Is Associated with Tumor Stage and Patient Survival

We screened DEGs between DTC and normal tissues from the microarray dataset by using 3 R packages. The general difference in gene expression is revealed in the volcano plot (Figure 1A–C). A total of 3 DEGs, namely PLA2R1, LYE1 and MMRN1, were significantly downregulated. As reported, PLA2R1 plays an important role in cancer suppression in a variety of cancers. We chose PLA2R1 as the main research focus.

PLA2R1 expression in DTC tissues was significantly lower than in normal tissues (Figure 1D), and the expression of PLA2R1 gradually decreased with increasing AJCC stage in DTC (Figure 1E,F). Although there was no significant association between the expression of PLA2R1 and the OS prognosis (*p* > 0.05), there was a trend towards an association of high expression of PLA2R1 with good prognosis (Figure 1G). However, low PLA2R1 expression was significantly correlated with poor PFI and DFS prognoses (*p* < 0.05) (Figure 1H,I).

### 3.2. Clinical Characteristics and PLA2R1 Expression in Clinical Specimens

Clinical specimens, specifically 54 tumor tissues and 61 para-cancerous tissues, were collected. Demographic characteristics and clinicopathological data are listed in Table 1. The current study included 33 female and 21 male DTC patients. The expression of PLA2R1 was lower in the tumor tissues than in para-cancerous tissues (χ^2^ = 37.0, *p* < 0.01) (Figure 2A–C, Table 2). PLA2R1 expression was much lower in more malignant tumors than in tissues from less malignant tumors (χ^2^ = 4.8, *p* < 0.05) (Table 1). Analysis of the data revealed that the ROC curve was in the upper left corner of the plot, with an AUC value of 0.966 (95% CI: 0.954–0.978), suggesting the high specificity and sensitivity of PLA2R1 in the diagnosis of DTC (Figure 2D).

### 3.3. Thyroid Cell Lines with Stable Overexpression or Silencing of PLA2R1 Were Established

To investigate the possible function of PLA2R1 in thyroid cancer cells, the expression of PLA2R1 was determined in the thyroid cancer cell lines BC-PAP, TPC-1, 8505c and FTC133, and in a normal thyroid cell line, using RT–PCR. As shown in Figure 3A, 8505c cells had a lower PLA2R1 expression, and FTC133 cells had a higher PLA2R1 expression than the other thyroid cancer cell lines. We selected these two cell lines for further experiments. Transduction of a lentiviral vector containing shRNA targeting PLA2R1 was used to generate shRNA-PLA2R1 cells (PLA2R1-silenced FTC133 cells). We then used the lentiviral overexpression system to generate PLA2R1-OE cells (PLA2R1-overexpressing 8505C cells) stably overexpressing PLA2R1. The normal thyroid cell line NTHY-ORI3-1 was used as the negative control. We successfully established thyroid cell lines with PLA2R1 knockdown and overexpression for further experiments.

### 3.4. PLA2R1 Inhibits Thyroid Cancer Cell Proliferation and Migration

As shown in Figure 3B,C, the cell proliferation ability in the shRNA-PLA2R1 groups was significantly higher than that in the negative control groups (670.56 ± 36.07 vs. 493.50 ± 35.59, *p* < 0.01), while PLA2R1 overexpression reduced the cell proliferation ability at 96 h (292.29 ± 5.07 vs. 376.29 ± 9.10, *p* < 0.01). As shown in Figure 3D,E, the overexpression of PLA2R1 inhibited cell migration (1.00 ± 0.04 vs. 0.62 ± 0.01, *p* < 0.01), while downregulation of PLA2R1 promoted this effect compared to the control group of thyroid cells at 48 h (2.22 ± 0.02 vs. 1.00 ± 0.13, *p* < 0.01). These results suggested that the overexpression of PLA2R1 suppresses thyroid cell proliferation and migration.

### 3.5. PLA2R1 Competes with FN1 to Interact with ITGB1

KEGG pathway analysis revealed that the main enriched pathways included ECM-receptor interaction and Focal adhesion (Figure 4A,B). However, integrin family members and fibronectin 1 (FN1) were differentially expressed in the three signaling pathways (Figure 4C). Finally, factors such as ITGA1, ITGB1, ITGB4 and FN1 were selected for further verification by qPCR, all of which were downregulated in PLA2R1-overexpressing cell lines (Figure 4D).

GEPIA was used to explore the relationships among the above factors, and the results showed that FN1 expression was negatively correlated with PLA2R1 expression (Figure 5A) and positively correlated with ITGB1 expression (Figure 5B) in thyroid cancer.

Co-IP experiments further clarified the interaction between PLA2R1, FN1 and ITGB1. The immunoprecipitation (IP) of ITGB1 was performed after the overexpression of PLA2R1, and the WB results indicated that FN1 was reduced (Figure 5C). PLA2R1 was found to be decreased after the IP of ITGB1, while FN1 was overexpressed (Figure 5D). The results showed that PLA2R1 competed with FN1 to interact with ITGB1.

### 3.6. PLA2R1 Regulates the ITGB1/FAK Axis and Inhibits EMT

Next, we explored the mechanisms by which PLA2R1 inhibits the proliferation and migration of DTC cells. The Focal adhesion (FAK) pathway, which regulates tumor progression, was identified in the KEGG pathway analysis. Therefore, we investigated whether the ITGB1/FAK pathway is affected by PLA2R1. Western blot analysis revealed that the upregulation of PLA2R1 reduced the protein levels of ITGB1 and FN1 (*p* < 0.001), while the downregulation of PLA2R1 increased the protein levels of FN1, ITGB1 and FAK (*p* < 0.001) (Figure 6A,B). EMT is defined as the transformation of epithelial cells into spindle cells with the loss of membrane E-cadherin expression and the gain of mesenchymal markers, such as vimentin, which promotes tumor initiation, progression and metastasis in human mammary epithelial cells. As shown in Figure 6, the results of Western blot analysis revealed that the overexpression of PLA2R1 decreased the levels of snail and slug, while the knockdown of PLA2R1 led to the opposite effect. It suggests that the overexpression of PLA2R1 could inhibit EMT in thyroid tumor.

The above effects of PLA2R1 on ITGB1, *p*-FAK, E-cadherin and caspase-3 levels were partially attenuated by FN1 treatment (Figure 6C). FN1 may function in thyroid cancer cells by interacting with ITGB1 and then activating the FAK signaling pathway.

### 3.7. The Effect of PLA2R1 on Tumorigenic Ability of Thyroid Cancer Cells Was Detected by the Tumor Formation Test

To further explore the effect of PLA2R1 on the tumorigenic ability of FTC133 cells, a nude mouse xenograft tumor model was established by the subcutaneous injection of FTC133 cells transfected with shRNA. Normal control group (NC group) and shRNAs (PLA2R1-shRNA group)-treated FTC133 cells were implanted subcutaneously into the axillae of nude mice. Tumor volumes were then measured daily and tumor growth curves were generated to compare the differences between the groups. Tumor volume and weight were measured after death and the differences were compared. The volume and size of transplanted tumors were significantly higher in the PLA2R1-shRNA group than in the NC group and were statistically different at day 22 (335.37 ± 127.02 vs. 151.23 ± 54.96, *p* < 0.05) and day 25 (546.91 ± 172.50 vs. 233.52 ± 94.78, *p* < 0.05) (Figure 7A). The weight of transplanted tumors in the PLA2R1-shRNA group was also higher than in the NC group; the difference was statistically significant (698.20 ± 149.00 vs. 216.20 ± 112.40, *p* < 0.001) (Figure 7A), and the results showed that the expression of PLA2R1 were decreased in PLA2R1-shRNA group. Moreover, after the knockdown of PLA2R1, the expressions of PLA2R1, ITGB1, FAK and N-cad increased (Figure 7B). In short, PLA2R1 knockdown promotes the tumorigenesis and proliferation of thyroid cancer via the FN1-mediated ITGB1/FAK axis in vivo.

## 4. Discussion

In this study, we evaluated the PLA2R1 expression in thyroid cancer tissues of different T grades through bioinformatics analysis and clinical sample analysis. The PLA2R1 expression level was negatively correlated with T grade, and low PLA2R1 expression was associated with a poorer prognosis. The above findings suggested that PLA2R1 could be a potential prognostic and therapeutic target. In the following study, it was found that the overexpression of PLA2R1 inhibited the proliferation and invasion of thyroid cancer cells, while the knockdown of PLA2R1 had the opposite effects. Furthermore, the in vivo tumorigenic assays revealed that the overexpression of PLA2R1 could inhibit the growth and proliferation of the thyroid tumor by modulating the ITGB1/FAK-signaling axis, compared to the control group in the animal study. In addition, the co-IP experiment and western blotting revealed that the overexpressed PLA2R1 competes with FN1 to bind ITGB1 and inhibit the activation of the downstream FAK-signaling pathway. By attenuating the interaction of FN1 with ITGB1, overexpressed PLA2R1 suppresses the activation of the downstream FAK-signaling pathway, which in turn influences the malignant proliferation of thyroid cancer cells. The above findings indicated an inhibitory role for PLA2R1 in the aggressive growth of thyroid cancer.

The influence of PLA2R1 on tumor cell function has been partially reported [16]. Studies have demonstrated that PLA2R1 is involved in several important biological processes in cancer, including triggering DNA damage, carcinogenesis, cell death and cell differentiation. Vindrieux et al. suggested that the knockdown of PLA2R1 increased the tumorigenicity of kidney cancer cells, supporting the role of PLA2R1 deficiency in promoting the growth of kidney cancer cells in vivo [12]. A study by Quach et al. indicated that PLA2R1 was a novel molecular target that could be used to control tumor growth and regulate the delivery of lipid-based nanodrugs [16]. Moreover, PLA2R1 promoter hypermethylation was found to be associated with aggressive subtypes of breast cancer, and PLA2R1 promoter hypermethylation was found to be a useful diagnostic and prognostic biomarker for breast cancer [17].

Despite the strong evidence supporting the role of PLA2R1 in cancer suppression, the mechanism of this role still needs to be further explored. It has been previously documented that PLA2R1 can inhibit cancer progression by activating the Janus kinase 2 (JAK2) pathway and inducing estrogen-related receptor alpha (ESRRA) [18,19]. This pathway affects mitochondrial function through the accumulation of reactive oxygen species (ROS), leading to senescence and apoptosis [20]. Another possible explanation is that PLA2R1 triggers DNA damage via the activation of the p53 signaling molecule, one of its downstream targets [21]. In contrast, little is known about the underlying mechanism by which PLA2R1 regulates tumor progression in thyroid cancer.

Our bioinformatic data clearly showed that PLA2R1 was closely associated with the ECM-receptor interaction, focal adhesion kinase (FAK) and PI3K-AKT-signaling pathways. The Integrin family, especially the core molecule ITGB1 and FN1, were differentially expressed in all the three signaling pathways. Integrins are laminin receptors that stabilize adhesion and promote signal transduction. Integrins generally act as receptors for downstream signaling, which in turn affect a variety of physiological processes, including cell differentiation, proliferation, adhesion, migration, autophagy, apoptosis and signal transduction [22,23,24,25]. It has been reported that integrin family proteins are closely involved in tumor proliferation and aggressive growth [26,27,28]. Moreover, integrins have been reported in the literature as potential receptors for FN1 [29], and upon this interaction, they have an important role in mediating tumor growth and metastasis [30,31,32].

In addition, we found that both FN1 and ITGB1 were enriched in the FAK-signaling pathway. FAK is a protein tyrosine kinase that is overexpressed and activated in many human cancers [33,34]. When FAK is activated by integrins, G protein-coupled receptor ligands or growth factors, FAK undergoes autophosphorylation, subsequently binding and activating downstream proteins, and ultimately leading to cell adhesion, migration, invasion, survival, proliferation, angiogenesis and the regulation of DNA damage repair [35,36]. Due to these effects and its overexpression in many cancers, FAK is associated with poor prognosis in some of these cancers [37,38,39]. Lihui et al. stated that microRNAs regulate the development and progression of gastric cancer through ITGB1/FAK signaling [40]. Research by Ryota et al. suggested that ITGB1-mediated FAK signaling is implicated in the formation of tumor vasculogenic mimicry-like networks, which may be a potential target for malignant cancer therapy [41]. Furthermore, Mitra et al. confirmed that fibronectin binding to ITGB1 could contribute to the recruitment and activation of proteins associated with signaling pathways, including the FAK signaling pathway [42]. Therefore, we assumed that the ITGB1/FAK axis is involved in tumor progression and distant metastasis in DTC.

In addition, according to the results of Co-IP, FN1 expression was negatively correlated with PLA2R1 expression and positively correlated with ITGB1 expression in thyroid cancer. Therefore, we hypothesized that the overexpressed PLA2R1 competes with FN1 to bind ITGB1 and inhibit the activation of the downstream FAK-signaling pathway by attenuating the interaction between FN1 and ITGB1, thereby affecting the malignant proliferation of thyroid cancer cells. It can be assumed that targeted regulation of PLA2R1 expression inhibits thyroid cancer progression through the FN1-mediated ITGB1/FAK axis.

However, it must be mentioned that our study is limited by the small sample size and single center data. Data from a larger population and multiple centers are needed to evaluate the expression and application of PLA2R1 in thyroid tumors.

## 5. Conclusions

In conclusion, the present study initially revealed that overexpression of PLA2R1 suppressed DTC progression by modulating FN1 expression through the ITGB1/FAK axis. Our findings suggest that PLA2R1 might be a prospective target for therapeutic strategies against DTC.

## Figures and Tables

**Figure 1 cancers-15-02720-f001:**
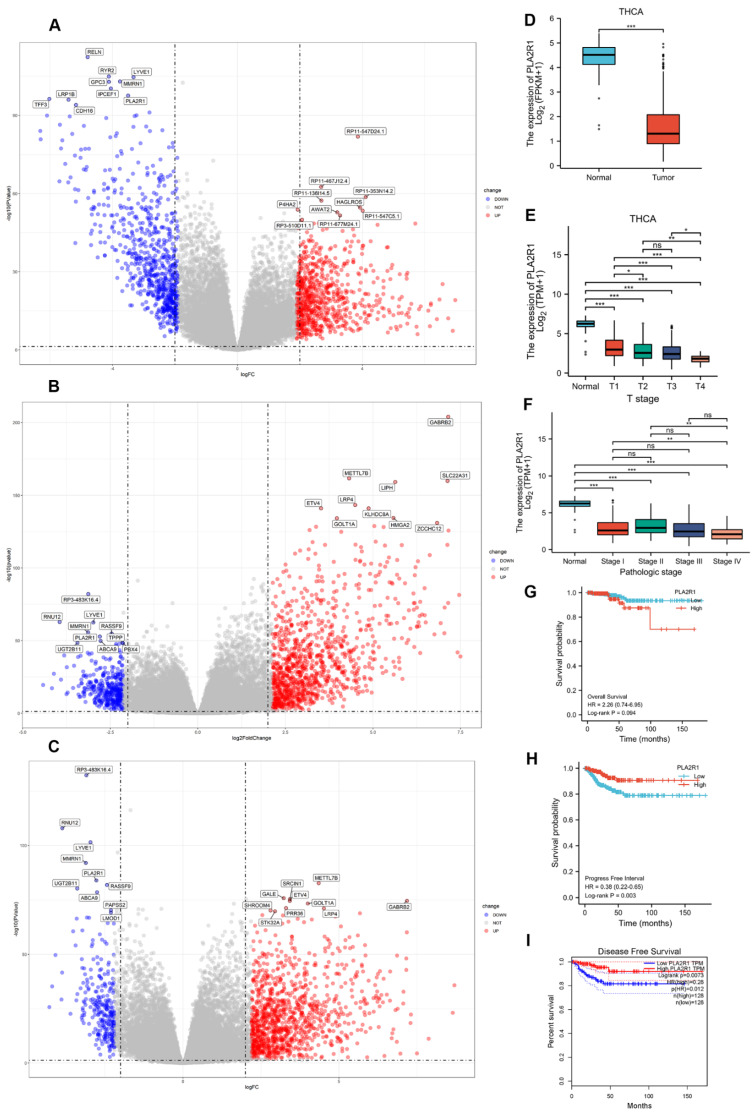
PLA2R1 expression and its correlation with prognosis in DTC by TCGA database analysis. (**A**–**C**) Volcano plot showing PLA2R1 gene expression in DTC data from TCGA as determined by three R packages. Log2 FC > 1 or log2 FC < −1 and FDR < 0.05 (or –log10 FDR > 1.3) were set as the thresholds for screening for differentially expressed genes (DEGs). FC, fold change; FDR, false discovery rate. (**D**–**F**) Expression of PLA2R1 in different T stages and pathological stages (TNM) of DTC. (**G**–**I**) Overall survival and progression-free survival were compared between the high and low PLA2R1 expression groups. ns, not significant; * *p* value <  0.05, ** *p* value  <  0.01, *** *p* value <  0.001.

**Figure 2 cancers-15-02720-f002:**
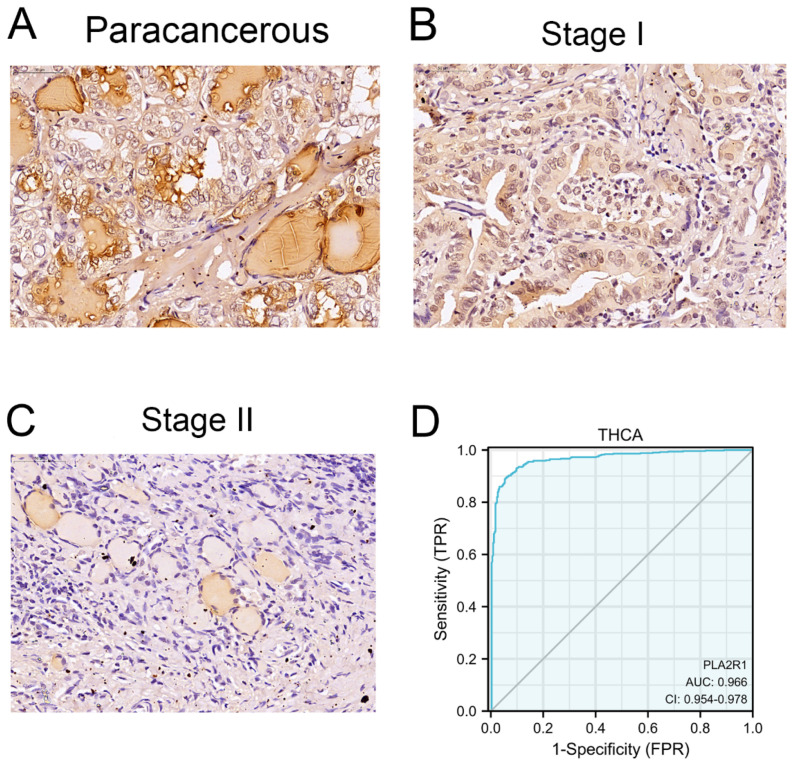
PLA2R1 expression in clinical specimens and its differential expression in tumors from different stages. (**A**) Immunohistochemistry was performed to detect PLA2R1 expression in Para-cancerous tissue. (**B**,**C**) PLA2R1 expression in thyroid cancer tissues with stage T1 and stage T2, as revealed by IHC staining. (**D**) ROC curves of PLA2R1 showed high sensitivity and specificity for the diagnosis of DTC. Scale bar, 25 μm.

**Figure 3 cancers-15-02720-f003:**
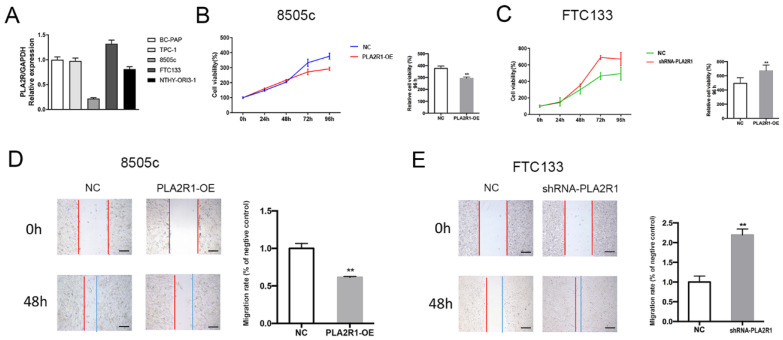
The effect of PLA2R1 on the behaviors of thyroid cell lines. (**A**) The gene expression of PLA2R1 in different thyroid cell lines including BC-PAP, TPC-1, 8505c, FTC133 and NTHY-ORI3-1. (**B**,**C**) Assessment of the proliferation of PLA2R1-OE cells and shRNA-PLA2R1 cells after plasmid transduction. (**D**,**E**) The effects of PLA2R1 on the migration of thyroid cell lines detected by wound healing assay at 48 h. Each experiment was repeated three times. PLA2R1-OE cells: PLA2R1-overexpressing 8505C cells; shRNA-PLA2R1 cells: PLA2R1-silenced FTC133 cells. ** *p* value < 0.01. Scale bar, 50 μm.

**Figure 4 cancers-15-02720-f004:**
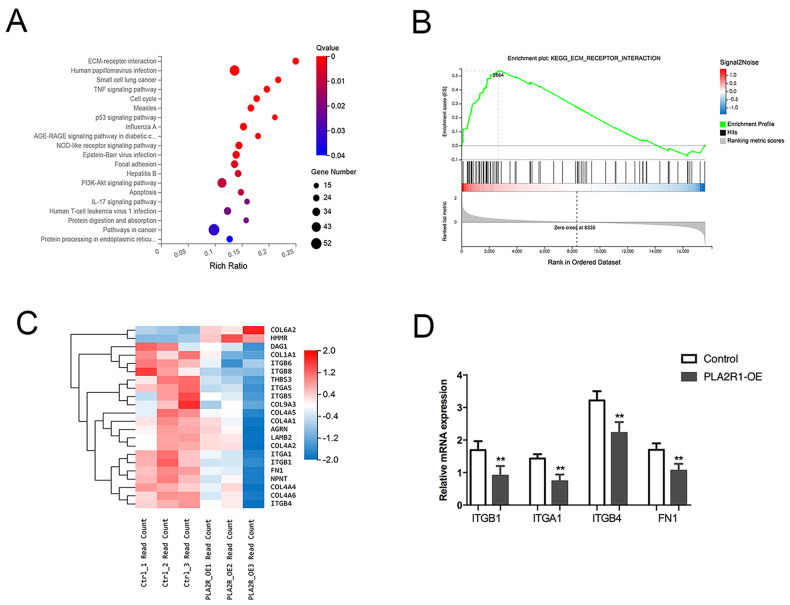
PLA2R1 affects the expression of extracellular matrix-associated molecules. (**A**) KEGG pathway analysis showed the top 20 enriched pathways. (**B**) Enrichment plot of ECM-receptor interaction pathway genes. (**C**) Heatmap of DEGs, mainly including the integrin family and FN1. (**D**) The relative mRNA expression of ITGB1, ITGA1, ITGB4 and FN1 after PLA2R1 overexpression. ** *p* value < 0.01.

**Figure 5 cancers-15-02720-f005:**
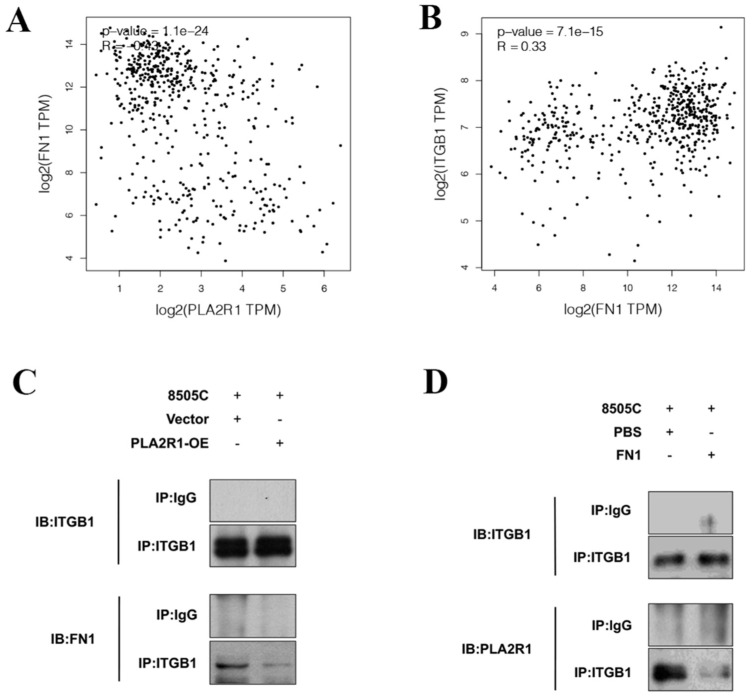
PLA2R1 compete with FN1 for binding to ITGB1. (**A**,**B**) Correlation analysis clarified the relationships of PLA2R1 expression with ITGB1 and FN1 expression in thyroid cancer. (**C**,**D**) Co-IP experiments validated the interactions between PLA2R1, FN1 and ITGB1. IP: immunoprecipitation; IB: immunoblotting. The uncropped blots are shown in Appendix A.

**Figure 6 cancers-15-02720-f006:**
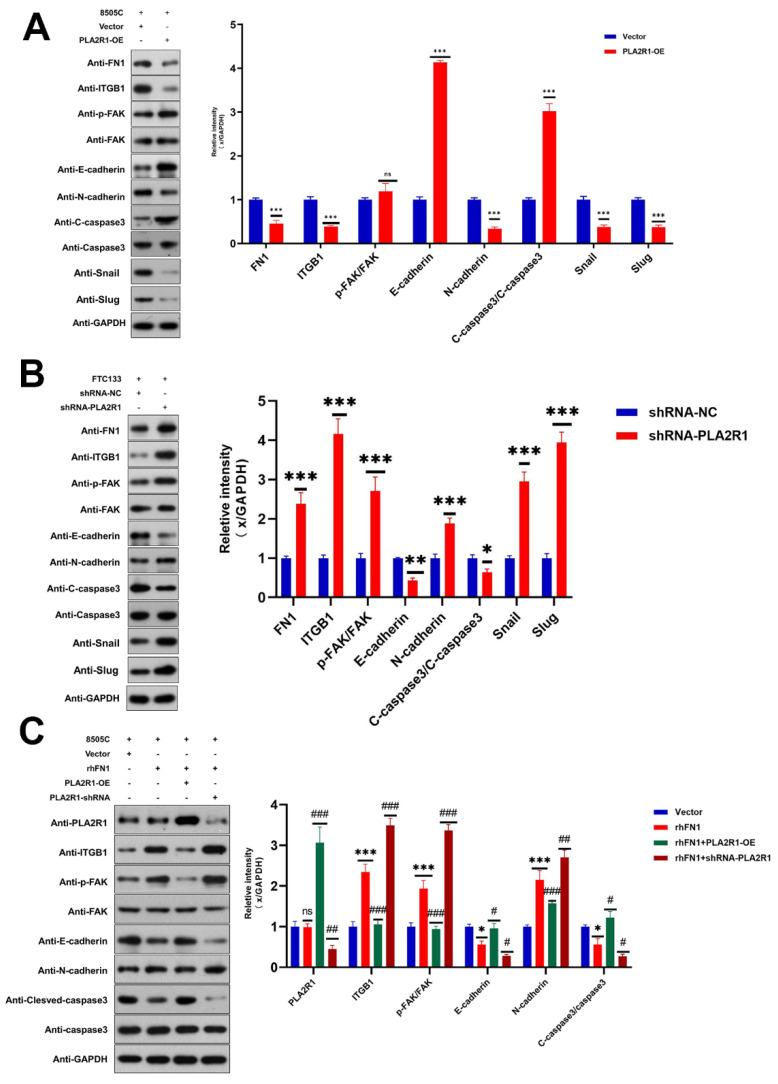
PLA2R1 suppresses DTC metastasis through the ITGB1/FAK axis. (**A**,**B**) Effects of PLA2R1 on the *p*-FAK, EMT protein, and apoptotic protein levels in thyroid cancer cell lines 8505c and FTC133, as determined by Western blotting. (**C**) FN1 (rhFN1) antagonizes the effects of PLA2R1 on ITGB1 expression and FAK phosphorylation in thyroid cancer cell lines 8505c and FTC133; ns, not Significant; * vs. Vector *p* < 0.05; ** vs. Vector *p* < 0.01; *** vs. Vector *p* < 0.001; # vs. rhFN1 *p* < 0.05; ## vs. rhFN1 *p* < 0.01; ### vs. rhFN1 *p* < 0.001. The uncropped blots are shown in Appendix A.

**Figure 7 cancers-15-02720-f007:**
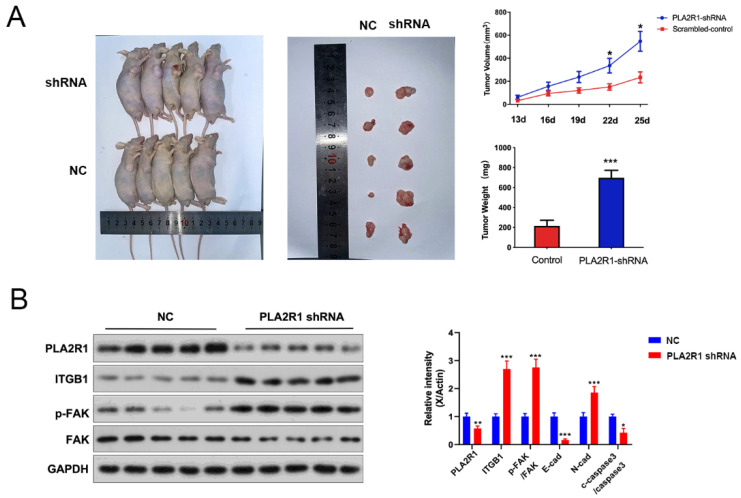
Knockdown of the PLA2R1 gene promotes the growth of thyroid cancer tumors. (**A**) The volume, size and weight of transplanted tumors in the scrambled group and shRNA-PLA2R1 group. (**B**) Detection of the expression of PLA2R1 and downstream-related genes in tumor tissues after shRNA-transfected FTC cells are tumor-bearing. NC: normal control. * *p* value <  0.05, ** *p* value  <  0.01, *** *p* value <  0.001. The uncropped blots are shown in Appendix A.

**Table 1 cancers-15-02720-t001:** Baseline Features of 54 DTC Patients.

Features	No. of Patients	PLA2R1 Expression		*p* Value
**All Patients**	**54**	**Low**	**High**	
		**32**	**22**	
**Age (years)**				0.724
≥55	23	13	10	
<55	31	19	12	
**Gender**				
Male	21	13	8	0.755
Female	33	19	14	
**T Infiltrate**				
T1	33	20	13	0.781
T2	19	11	8	
T3	2	1	1	
**Lymphatic Metastasis (N)**				
N0	26	16	10	0.867
N1	27	16	11	
**AJCC stage**				
I	44	23	21	0.030
II	10	9	1	
**Tumor size**	54	32	22	0.790

**Table 2 cancers-15-02720-t002:** The results of sign test analysis.

PLA2R1	Tumor Tissue		Para-Cancerous Tissue		*p* Value
**Expression**	**Cases**	**Percentage**	**Cases**	**Percentage**	**0.0000**
Low	32	59.3%	4	5.6%	
High	22	40.7%	57	93.4%	

## Data Availability

Data are contained within the article and Appendix A.

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
