# Peer review of "PLA2R1 Inhibits Differentiated Thyroid Cancer Proliferation and Migration via the FN1-Mediated ITGB1/FAK Axis"

_cancers, 2023, doi:10.3390/cancers15102720_

Round 1
Reviewer 1 Report
A frequent malignant tumor with an increasing incidence worldwide is thyroid cancer. Papillary thyroid cancer (PTC), follicular thyroid cancer (FTC), anaplastic thyroid cancer (ATC), and medullary thyroid cancer are the four most prevalent types of thyroid cancer, which is a frequent endocrine carcinoma with a high prevalence. The most prevalent thyroid cancer subtypes are DTC. Although there are conventional thyroid cancer treatments, they have several drawbacks and considerably shorten patient survival periods. Thus, it's crucial to discover prognostic factors and look into the mechanism driving DTC's distant metastases. In this paper, the authors explain how PLA2R1 contributes to thyroid cancer. PLA2R1 is closely associated with the ECM-receptor interaction, focal adhesion kinase (FAK), and PI3K-AKT signaling pathways. They demonstrated that overexpressing PLA2R1 greatly reduced the proliferation and migration of thyroid cancer cells and that silencing of PLA2R1 partially inhibited both effects. However, I have a few concerns.
What are the patient sample inclusion and exclusion criteria?
What is the scale bar used in IHC and migration assay images?
In figure legends, it is unclear which cell line is used for each experiment.
The time period used for the western blots' images is not clear.
A model should be provided to allow the mechanism to be better understood.
Apart from the fact that the authors use in vitro cell lines to illustrate the mechanism. Yet, this experiment is relatively easy. If the authors showed this mechanism in the in vivo model, that would be interesting.
Reviewer 2 Report
General comments:
In this manuscript, Zheng et al. have investigated the role of PLA2R1 in differentiated thyroid cancer (DTC) referring to it as a novel target for therapeutic strategy against DTC.
After an accurate re-reading of the paper, I suggest (see comments in “Major points”) improving the functional results in order to speculate PLA2R1 as a promising marker and novel therapeutic target for thyroid cancer.
In fact, some integrations and suggestions for improving the manuscript are provided below as specific comments.
Minor points:
· The legend of Table 1 reports a wrong capture for Lymph Node Positive, maybe 0 and 1 may be replaced with =0 and >0 as described in the table.
· According to the figures, my recommendation is to increase the quality, especially for the volcano plots shown in Figure 1, for example, the legends of the axes and the highlighted DEGs cannot be read well.
Major points:
· Is there a connection between PLA2R1and FN1 in the thyroid cell models? Do you think your analysis reveals something specific for this tumor or has your study confirmed and supported what is already known about other types?
· Western blots data quality is very poor in terms of data presentation. I suggest including the whole western blot showing all bands and molecular weight markers in the Supplementary Materials. Moreover, Figures 5 and 6 show only the qualitative image of western blots, can the authors report and describe the densitometry readings/intensity level of each band? In this regard, how significantly the ITGB1/FAK axis was affected by PLA2R1 (Figure 6)? I mean that the figure is lack quantitative data referring to the intensity level of the protein bands corroborated by the statistical analysis.
· I think that the data shown in this manuscript are very interesting and should be useful to clarify the role of PLA2R1 in thyroid cancer, but it is not clear to me how the authors come up with the hypothesis that PLA2R1 suppresses DTC metastasis through the ITGB1/FAK axis. Any deeper functional validation of the data is missing. In my opinion, a stronger mechanistic insight would be required.
Author Response
Plesase see the attachment.

Reviewer 3 Report
In the manuscript ”PLA2R1 inhibits differentiated thyroid cancer proliferation and migration via the FN1-mediated ITGB1/FAK axis”, the authors demonstrated that the expression of PLA2R1 was reduced in thyroid cancer tissues, and are correlated with prognosis. Further, the author suggested that the underlining mechanism is due to the competition of PLA2R1 and FN1 in the binding to ITGB1.
Overall, most of the study is solidly designed and well executed. The language is overall clear; however, redundancies, grammatical errors, and typos were seen sporadically. The terms, especially tumor staging related, are sometimes confused. The authors need to polish the language and revise the typos in revision. The authors should have an in-depth discussion on the limitations and some inconsistencies in the study. The methodology is overall solid. Most of the figures are well-made and properly labeled.
Here are the major points the author needs to revise or clarify:
1. The inconsistency and misleading between table 1 and figure 1. The author needs to use consistent terms/staging here. For example, the authors used AJCC/TNM staging but did not provide M. And then used the “pathologic stage” in figure 1F. The author should show consistent information/terms between table 1 and figure 1.
2. According to the latest ACJJ staging system and the American cancer society, 55 years old, not 45 years old is used as a threshold for staging. Why the author used 45 year old here? The author should re-group the patient samples and revise table 1 and figure 1. Or have an explanation.
3. In table 1, the author showed 44 patients are AJCC stage I and 10 patients are AJCC stage II. And described in the text that “the expression of PLA2R1 gradually decreased with increasing AJCC stage in DTC (Figure 1E, F).” So where is the T3 and T4 stage come from? Were you referring to the T as the Tumor from TNM staging here? If yes, why don’t just use TNM staging in table 1?
4. In figure 1E, it seems the authors had excluded the “outlier” samples from the normal group, T2, and T3 group. The authors need to provide statistical evidence to show the rationale. Cause clearly if the data include the “outlier” samples, the conclusion and significance may change.
5. In figure 3D&E, apparently this is a wound healing assay. And I assume the bar graph quantifies the wound healing assay. Why the authors described it as Boyden chamber-based migration assay in method 2.9???
6. In figure 4C, previous reports showed that high HMMR promotes migration. The PLA2R overexpression increases the HMMR level. The authors need to have an explanation.
7. In figure 6, the authors should check the expression of EMT-related TFs, e.g. nuclear-b-catenin, snail, slug, zeb1, zeb2. Also, ITGB/FAK axis is related to tumor invasion and the elevated expression of invasion-related genes, e.g., MMP9. The author should check the tumor invasion ability with PLA2R-OE and shRNA-PLA2R.
8. Overall, the authors were trying to prove PLA2R is associated with migration. However, in table 1, the author showed that PLA2R expression is not correlated with lymphatic invasion or metastasis. The data shows that PLA2R is more likely to be correlated with prognosis and apoptosis. The authors need to have an in-depth discussion. Or provide some in vivo data in animal xenograft/genetic models.
Minor:
9. In table 1, are there any reasons why the author decided not to just use >45 yrs and < 45 yrs, but use 0 and 1 and then explain it? And please don’t use “less than” or “more than”45 years. It’s under or above.
10. Figure 1A is unreadable, please enlarge the font size.
11. In the bar graph of figure 3D and E, the authors should use “% cell-free area” or “% wound healed”, instead of “relative cell migration”. Please refer to other publications.
12. The authors should try avoiding using the word “might”, e.g. the figure legend of figure 5.
13. Please refine the language and use consistence terms. e.g. tumor and tumour.
Round 2
Reviewer 2 Report
The authors have made substantial changes to this article in response to the reviewer’s comments, including some additional information and results by adding animal experiments too. WB quantitative data are very beneficial to better visualize and clarify the obtained results in the study. The in vivo tumorigenic assays gave added value to functional mechanistic insight because they better validate the obtained results corroborating the hypothesis that PLA2R1 inhibits the growth and proliferation of thyroid tumors through modulating the ITGB1/FAK signaling axis. However, my recommendation is always to increase the Volcano Plots quality prior to manuscript publication in accordance with the journal’s requirements.
Author Response
Response to reviewer 2:
The authors have made substantial changes to this article in response to the reviewer’s comments, including some additional information and results by adding animal experiments too. WB quantitative data are very beneficial to better visualize and clarify the obtained results in the study. The in vivo tumorigenic assays gave added value to functional mechanistic insight because they better validate the obtained results corroborating the hypothesis that PLA2R1 inhibits the growth and proliferation of thyroid tumors through modulating the ITGB1/FAK signaling axis. However, my recommendation is always to increase the Volcano Plots quality prior to manuscript publication in accordance with the journal’s requirements.
Response: Thank you for your comments and we think it is very important for us. We have improved the quality of the volcano plots in accordance with the magazine's requirements.
Reviewer 3 Report
Thank you for the responses from the author. The revision addressed most of my concerns.
I have a few minor suggestions:
1. For method 2.10, please use "subcutaneous xenograft mouse model".
2. Method 2.11. Statistical analysis should be 2.12.
3.Result 3.6. I suggest describing caspase-3 as an apoptotic factor, not EMT factor. Normally, caspase-3 was not considered an EMT modulator. The most used key EMT factors or TFs are snail, slug, zeb1, zeb2, twist1, and twist2. I still suggest the authors provide the WB for any of these EMT-TFs. This could strengthen the conclusion on EMT inhibition.
Author Response
Response to reviewer 3:
Thank you for the responses from the author. The revision addressed most of my concerns.
I have a few minor suggestions:
Thank you for your comments and suggestions, we carefully examined our manuscripts. To address the concerns raised by reviewers, we have performed the requested additional experiments. We have revised the manuscript with track changes mode in MS Word and responded to your comments point-by-point at the bottom of this letter.
- For method 2.10, please use "subcutaneous xenograft mouse model".
Response: Thank you for pointing out that. We have modified method 2.10 to "subcutaneous xenograft mouse model".
- Method 2.11. Statistical analysis should be 2.12.
Response: We are very appreciative for your comments. We have made changes in response to your comments in method 2.12.
3.Result 3.6. I suggest describing caspase-3 as an apoptotic factor, not EMT factor. Normally, caspase-3 was not considered an EMT modulator. The most used key EMT factors or TFs are snail, slug, zeb1, zeb2, twist1, and twist2. I still suggest the authors provide the WB for any of these EMT-TFs. This could strengthen the conclusion on EMT inhibition.
Response: Thank you for your comments and we agree that the suggestion is professional, instructive and of great significance to our paper. The western blot assay for snail and slug has been supplemented in the text to validate our results. Based on the results of supplementary WB assay, we can see that the overexpression of PLA2R1 inhibits EMT in thyroid cancer, whereas knockdown of PLA2R1 has the opposite effect.